

# Deposition freezing, pore condensation freezing and adsorption: three processes one description?

Mária Lbadaoui-Darvas[1], Ari Laaksonen[2,3], and Athanasios Nenes[1,4]

[1]Laboratory of Atmospheric Processes and their Impacts, School of Architecture, Civil and Environmental Engineering, École Polytechnique Federale de Lausanne, 1015 Lausanne, Switzerland
[2]Finnish Meteorological Institute, 00101 Helsinki, Finland
[3]Department of Applied Physics, University of Eastern Finland, 70211 Kuopio, Finland
[4]Institute of Chemical Engineering Sciences, Foundation for Research and Technology Hellas (FORTH/ICE-HT), 26504 Patras, Greece

**Correspondence:** Ari Laaksonen (ari.laaksonen@fmi.fi), Athanasios Nenes (athanasios.nenes@epfl.ch)

**Abstract.** Heterogeneous ice nucleation impacts the hydrological cycle and climate through affecting cloud microphyiscal state and radiative properties. Despite decades of research, a quantitative description and understanding of heterogeneous ice nucleation remains elusive. Parameterizations are either fully empirical or heavily rely on classical nucleation theory (CNT), which does not consider molecular level properties of the ice nucleating particles - which can alter ice nucleation rates by
orders of magnitude through impacting pre-critical stages of ice nucleation. The Adsorption Nucleation Theory (ANT) of heterogeneous droplet nucleation has the potential to remedy this caveat and provide quantitative expressions in particular for heterogeneous freezing in the deposition mode (the existence of which has even been questioned recently). In this paper we use molecular simulations to understand the mechanism of deposition freezing and compare it with pore condensation freezing and adsorption. We put forward the plausibility of extending the ANT framework to ice nucleation (using black carbon as a case
study) based on the following findings: i) The quasi-liquid layer at the free surface of the adsorbed droplet remains practically intact throughout the entire adsorption and freezing process, therefore the attachment of further water vapor to the growing ice particles occurs through a disordered phase, similar to liquid water adsorption. ii) The interaction energies that determine the input parameters of ANT (the parameters of the adsorption isotherm) are not strongly impacted by the phase state of the adsorbed phase. Thus, not only the extension of ANT to the treatment of ice nucleation is possible, but the input parameters
are also potentially transferable across phase states of the nucleating phase.

# 1   Introduction

Heterogeneous nucleation is a major of source of cloud droplets and ice particles (Hoose and Möhler, 2012; Kanji et al., 2017), whose number, size and relative amounts drive the evolution of clouds, their radiative properties and impacts on the
hydrological cycle and climate (Fowler and Randall, 1996; Pinto, 1998). Its importance has been known for almost a century;





yet despite countless studies on the topic, the mechanism of heterogeneous nucleation remains poorly understood (Pruppacher et al., 1998; Laaksonen and Malila, 2021). To date there is no established theory of heterogeneous nucleation of ice and droplets on insoluble particles. The lack of a such a functional theoretical framework impedes a full understanding of aerosol–cloud interactions, which constitute one of the most uncertain drivers of anthropogenic climate change (Boucher et al., 2013; Seinfeld
et al., 2016).

The oldest and most commonly used theory of heterogeneous nucleation is the "classical nucleation theory" (CNT) (Fletcher, 1958). CNT determines the conditions required for the formation of a critical droplet/ice particle on a surface from a metastable phase, such as supersaturated vapor or supercooled liquid, assuming that it occurs in a single step without any considerations of pre-critical interactions between the surface and the vapor/liquid phase, and assuming that the energetic interactions of the
molecules near the surface are the same as in the bulk. Because of these simplifications, CNT does predict qualitatively the nucleation rate trends, but can lead to very large biases (DeMott et al., 2010), which are partially mitigated by changing the value of the interaction parameters to fit observations. In reality, considerable water adsorption on the surface of insoluble particles occurs already under subsaturated conditions (RH<100 %), and the presence of pre-critical adsorbed water impacts the nucleation process because much of the water required to generate a critical cluster, and their associated energy content, is
available even before activation. Furthermore, the heat of adsorption changes as a function of adsorption layer thickness (Hill, 1949), which might also impact heterogeneous nucleation.

Atmospherically relevant insoluble particles are not always perfectly wettable, in which case adsorbed water exists as patches or clusters. Heterogeneous ice nucleation is thought to occur on so called ice nucleation active sites (INAS) whose nanoscale properties - both chemical and structural - enhance freezing of adsorbed clusters , whereas other locations that collect pre-
critical clusters might have an opposite effect. For instance, ice nucleation is suppressed on molecularly rough or curved graphene (Lupi et al., 2014a), in wedge-shaped pores of black carbon (Bi et al., 2017) or AgI (Roudsari et al., 2022) for specific wedge angles which block the formation of initial ice embryos. Small steps and edges also alter IN activity of multiple surfaces (Roudsari et al., 2022). For ice-nucleating proteins, both their length and the lateral distance between the aggregate chains alter ice nucleation rates (Qiu et al., 2019), the latter in a non-monotonous way. Certain wedge angles in graphene (Bi et al., 2017)
and lateral distances in protein aggregates (Qiu et al., 2019) can reduce ice nucleation rates up to 8 orders of magnitude or reduce the freezing temperature by 10-15 °C. Therefore neglecting these features may introduce non-negligible bias into ice nucleation parameterizations. However, neither the wedge angle nor the aggregation state dependence of immersion freezing rates could be predicted from CNT, because these alterations of the ice nucleation rate arises from pre-critical surface-water interactions.

A theoretical framework that promises to treat the above described caveat of CNT is the adsorption nucleation theory (ANT) (Sorjamaa and Laaksonen, 2007; Laaksonen, 2015; Laaksonen and Malila, 2016). It has the potential to remedy the problem of neglecting pre-critical surface-water interactions by introducing multilayer adsorption in the model. Currently ANT-based parameterizations exist for the cloud condensation (CCN) activity of the most common insoluble atmospheric particles: dust (Kumar et al., 2009a, 2011), black carbon (Kumar et al., 2009b; Laaksonen et al., 2020) and volcanic ash (Lathem et al.,
2011). ANT-based droplet parameterizations have been implemented regional (Bangert et al., 2012) and global climate models





(Kumar et al., 2009b; Karydis et al., 2012), in particular climate model of NASA Global Modeling Initiative (GMI), GEOS-Chem (Karydis et al., 2012, 2017) EC-Earth () and NorESM (). ANT uses the Frenkel-Halsey-Hill (FHH) (Frenkel, 1947; Halsey, 1948; Hill, 1952) adsorption isotherm to express the equilibrium supersaturation ($S$) over a growing droplet as:

$$ln(S) = -\frac{A_{FHH}}{N_d^{B_{FHH}}} + \frac{2\gamma\nu_w}{k_B Tr}. \tag{1}$$

In the second (Kelvin) term of Equation 1 $r$ is the droplet radius $\gamma$ is the surface tension, $\nu_w$ is the volume of a water molecule and $k_B$ stands for the Boltzmann constant. The first term is the FHH isotherm (Halsey, 1948) expressed as a function of the number of adsorbed monolayers of water, $N_d = \delta/\delta_m$, where $\delta$ and $\delta_m$ denote the average water layer thickness and the thickness of a monolayer respectively. $A_{FHH}$ and $B_{FHH}$ are the isotherm parameters, which describe the energetics of adsorption and at the macroscopic scale are related to the lateral spread and the width of the adsorbed water layer. $A_{FHH}$

represents the interaction energy of the first adsorbed layer of water with the surface together with the lateral interactions within the layer. $B_{FHH}$ is related to how rapidly the interaction energy between the surface and the consecutive layers decays. $A_{FHH}$ and $B_{FHH}$ are directly linked to intermolecular interactions between the adsorbent and the adsorbate. Since the interaction energies that define $A_{FHH}$ and $B_{FHH}$ directly depend on the nanoscale properties of the surface, ANT provides a macroscopic framework of water vapor adsorption and CCN activation with direct links with the molecular-scale interactions.

ANT has been suggested to be suitable for modeling ice nucleation in the deposition mode. Deposition freezing occurs when an ice nucleating particle (INP) is in contact with water vapor saturated with respect to ice, which is thought to be the major source of cloud ice particles in the colder regions of mixed phase clouds as well as in cirrus clouds (Hoose and Möhler, 2012). Deposition-mode freezing parameterizations rely on CNT (Hoose et al., 2010; Niedermeier et al., 2011; Savre and Ekman, 2015; David et al., 2019) or on empirical approaches (Meyers et al., 1992; Ullrich et al., 2017) that neglect the

impact of pre-critical surface-water interactions. Additionally, the microphysical mechanism of deposition freezing is not fully understood. While the majority of researchers agree about the importance of deposition freezing as a primary ice formation mechanism in clouds, an extensive set of studies (Christenson, 2013; Marcolli, 2014; Wagner et al., 2016; Marcolli, 2017; Mahrt et al., 2018; Nichman et al., 2019; David et al., 2019; Marcolli et al., 2021) have questioned its existence alltogether, as freezing experiments on porous dust (illite, kaolinite and Arizona Test Dust) (Marcolli, 2014; Wagner et al., 2016; David

et al., 2019) and soot (Mahrt et al., 2018; Nichman et al., 2019; Marcolli et al., 2021) particles together with scanning electron microscopy and molecular simulations (David et al., 2019) show that deposition freezing occurs on surfaces with nanopores whose diameter allows for the formation of a critical ice embryo but is small enough to collect and hold confined pore water via the inverse Kelvin effect. Based on the result of these studies a hypothesis has been built which states that what is observed as deposition freezing is in reality pore condensation freezing (PCF), a mechanism which involves the filling of nanopores with

supercooled water due to capillary condensation and the subsequent immersion or homogeneous freezing within the water-filled pore causing out-of-pore growth of ice. Molecular simulations in the above set of studies (David et al., 2019) support this assumption, but they use pores that are prefilled with ice in the initial configuration and therefore the initial steps of pore filling and freezing within the pore are not explicitly resolved. From the experimentally observed temperature dependence of IN efficiency of porous aerosol particles, the authors conclude that PCF below 235 K involves homogeneous freezing, however



the ice nucleation experiments that lead to this conclusion are performed in droplet emulsions or aerosolized particles, and freezing in pores is not directly observed.

Indirect evidence for the potential extension of ANT to ice nucleation comes from both molecular simulations and experiments. A molecular dynamics study of water droplets on graphite and graphene oxide surfaces with variable hydrophilicity (Lupi and Molinero, 2014) showed that interfacial water in the first and second adsorption layers forms hexagonal arrangements similar to those of the basal plane of hexagonal ice (at 220 K and room temperature). Similar findings were reported in immersion and contact freezing simulations on graphene (Lupi et al., 2014a). Yang et. al. (Yang et al., 2021) used NEXAFS spectroscopy to elucidate the structure of adsorbed water on AgI and found that water tends to form ice-like structures in adsorption layers even at room temperature, further supporting the similarity of the equilibrium structure of adsorbed liquid water and ice (and hence their corresponding FHH parameters). These findings suggest that the values of $A_{FHH}$ and $B_{FHH}$ can be the same for water and ice. Raman spectroscopy (Kahan et al., 2007), atomic force microscopy (Gelman Constantin et al., 2018) as well as molecular simulations (Neshyba et al., 2009; Hudait et al., 2017; Pickering et al., 2018) evidence the formation of a quasi-liquid layer at the solid/vapor interface of ice, which implies that the mechanism of water adsorption and deposition freezing are qualitatively similar (i.e., the attachment of a water molecule to a liquid adsorption layer and a frozen one both occur via interactions with a disordered surface layer of water). Although concerns have been raised about the ability of the models used for the molecular simulations to reproduce properties of the graphene/water interface (Qiu et al., 2018) , the majority of the observational and modeling studies to date open the possibility of using ANT as a unified theory of heterogeneous droplet nucleation and deposition freezing. To rigorously demonstrate the plausibility of the extension of ANT to deposition freezing, one has to i) provide a complete description of the mechanism of deposition freezing at a molecular scale from the initial stages of water vapor adsorption to the nucleation of ice, ii) highlight similarities and differences between frozen and liquid adsorption layers structure, and, iii) compare interaction energies included in $A_{FHH}$ and $B_{FHH}$ for liquid water and ice.

In this work we address all of the above three crucial points using atomistic and coarse-grained molecular simulations. Molecular simulations have long been employed to study immersion freezing for all atmospherically-relevant substrates, e.g.: (Lupi et al., 2014a, b; Bi et al., 2017; Qiu et al., 2018, 2019; Sosso et al., 2016a, b). These studies used a variety of enhanced sampling techniques that allow for estimating ice nucleation rates using the assumptions of CNT (Pedevilla et al., 2018) or from the liquid-to-ice flux (Bi et al., 2017; Hussain and Haji-Akbari, 2021). Grand Canonical Monte Carlo simulations are widely used to study adsorption and even to quantitatively reconstruct adsorption isotherms (Lbadaoui-Darvas et al., 2021), however the purely stochastic approach only allows for understanding equilibrium properties of the adsorbed layer and lacks information about adsorption dynamics, which are crucial for understanding the freezing process. David et. al. remedied this caveat by adapting hybrid Grand Canonical Monte Carlo/molecular dynamics (GCMC/MD) simulations to study deposition freezing on flat and porous silica surfaces (David et al., 2019). However, as mentioned in the previous sections, the initial structures of these simulations consisted of porous systems pre-filled with hexagonal ice, therefore it addressed out of pore ice growth but not pore filling and in-pore freezing. In this study we remove this constraint and extend the analysis to the initial steps of the adsorption/freezing process that occur before pore-filling is complete. The GCMC/MD simulations, that are performed at a





coarse grained resolution to allow for the observation of the otherwise slow freezing processes, are complemented by atomistic scale molecular dynamics simulations, that are aimed at providing a quantitative assessment of the similarities between the energetic background of adsorption and deposition freezing. The two complementary sets of simulations clarify the extent to which adsorption, deposition freezing and PCF can be described using a unifying theory based on ANT valid for both deposition mode freezing and droplet nucleation. For this study we chose multilayer graphene as a model surface, given the

potentially important role of soot - mostly from aircraft emissions - as an INP in the upper troposphere (Hoose and Möhler, 2012; Kanji et al., 2017) and the lack of significant knowledge to date on the subject. Soot is active in deposition mode freezing (Hoose and Möhler, 2012). Laboratory experiments have shown that at temperatures below about 235 K, deposition ice nucleation occurs on porous black carbon particles at considerably lower critical supersaturations than on non-porous black carbon particles (DeMott et al., 1999; Möhler et al., 2005; Mahrt et al., 2018; Nichman et al., 2019). ANT parameterizations

of droplet nucleation on various type soot are available (Laaksonen et al., 2020).

## 2    Methods

### 2.1    Grand Canonical Molecular Dynamics

GCMC/MD simulation were performed using the LAMMPS molecular simulations program package (Thompson et al., 2022). The simulation cell initially contained a graphite slab consisting of 9 layers with $8 \times 8$ nm lateral dimensions in contact with an

empty headspace of $8 \times 8 \times 7$ nm. A small ice seed consisting of $\sim 30$ water molecules was placed in the pore with the hexagonal plane in contact with the surface in order to increase the computational efficiency by inducing initial adsorption of the water molecules. The size of the ice seed was chosen to be much smaller than the size of a critical ice nucleus (100-150 molecules) determined from molecular simulations of immersion and contact freezing on graphitic surfaces (Lupi et al., 2014a) to ensure that the seed will not induce ice nucleation as an artefact. The effect of the seed on ice nucleation is discussed in the light of the

results. The target vapor pressure (chemical potential) of water was set to a value that corresponds to a 300% supersaturation with respect to ice ($S_i = 3$) at 200 K. The target vapor pressure is not to be confused with the instantaneous vapor pressure in the simulation box, it is strictly the vapor pressure that corresponds to the adsorption layer structure in the converged simulation. The actual vapor pressure in the initial steps of the simulation, while the number of water molecules shows an increasing trend, increases upon the addition of water molecules and converges to the target pressure. The simulation temperature was

kept constant using the Nosé-Hoover thermostat (Evans and Holian, 1985). Water molecules were modelled using the coarse-grained mW water model (Molinero and Moore, 2009). The water-carbon interaction parameters $\epsilon_{CW}$=0.54392 kJmol$^{-1}$, $\sigma_{CW}$=0.32 nm were adapted from (Lupi et al., 2014b) to reproduce the experimental contact angle of water (86°) at 300 K.The graphitic slab was kept frozen during the simulation, thus no potential was necessary to describe interactions between the carbon atoms.

Given that the objective of this simulation was to observe the initial steps of deposition freezing, the run was stopped when an equilibrium number of water molecules was reached (at around 8 nanoseconds). In every Monte Carlo step, additions and deletions of water molecules were attempted and accepted or rejected according to the Metropolis algorithm until a stable





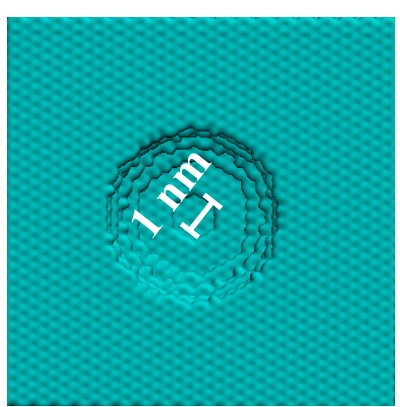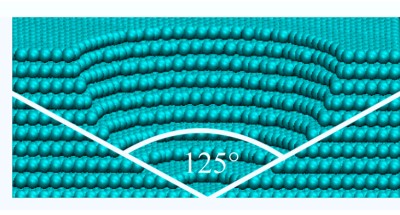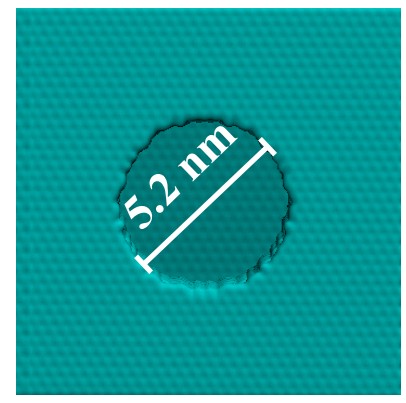

**Figure 1.** a) Snapshots of the pore geometries. Top and side views of the hemispherical pore and top view of the cylindrical pore.

number of water molecules was reached. Metropolis sampling of displacement moves was disabled so as to ensure that the molecular structure of the adsorbed droplets is solely a result of deterministic dynamics. Five Monte Carlo insertion/deletion

steps were performed for each classical molecular dynamics steps with an integration timestep of 5 fs. The MC/MD ratio of 5 leads to a relatively slow convergence of the vapor pressure, however it ensures a more efficient equilibration of the regions affected by insertion or deletion, thus giving a more exact description of the instantaneous structure of the adsorbed droplets. To check the dependence of the mechanism on this parameter, the first 2.5 nanoseconds of the simulation were repeated using MC/MD ratio of 20 (see Appendix A for the results); the MC/MD ratio numerically (but not qualitatively) impacts the observed

freezing curves - but not any conclusions regarding the mechanism of deposition freezing. Ice formation along the trajectory was analysed using the CHILL+ algorithm (Nguyen and Molinero, 2015) implemented in the OVITO software (Stukowski, 2009). To understand how surface porosity impacts the mechanism of deposition freezing, we repeated the simulations using surfaces containing a cylindrical and a hemispherical pore with a radius of 2.6 nm (measured at the top of the pore). Figure 4 shows the pore geometries. The depth of the pores is 2.7 nm.

While graphite is a good choice for an exploratory molecular simulations study as optimal interaction parameters that reproduce the contact angle of water on graphitic carbon, have been reported for both atomistic (Werder et al., 2003; Sergi et al., 2012) and coarse grained water models (Lupi et al., 2014a; Lupi and Molinero, 2014; Lupi et al., 2014b; Qiu et al., 2018), we note that the coarse grained model used in our simulation overestimates the ice nucleation ability of graphite because it can only reproduce the free energy of the water/graphite interface and not that of the ice/graphene interface because of its

resolution. This implies that the results of the GCMC/MD simulations are not necessarily indicative for real graphitic carbon (Qiu et al., 2018). The system models graphene structure with a surface that favors ice nucleation because because of the negative free energy of the ice/graphite interface, i.e. a surface that has the lattice of graphite but a lot lower ice/surface binding free energy. Because of the above caveat we refrain from any estimating quantitative data from the coarse grained simulation, but we maintain that the model is still able to provide a qualitative picture of ice nucleation and pore condensation valid for a

hypothetical surface with graphene structure.



## 2.2 Atomistic molecular dynamics simulations

To avoid bias coming from the above limitation of the coarse grained model and to be able to explicitly analyse hydrogen bonding, the interaction energies were calculated from atomistic molecular dynamics simulations performed using GROMACS (Abraham et al., 2015). The systems were built up of a graphite slab consisting of 10 layers brought manually in contact with a pre-equilibrated ice and water slab having the same lateral dimensions, consisting of 2880 water molecules. The ice slab was purely hexagonal and the contact between the graphite and the ice was established with the basal hexagonal (0001) plane of the latter. The graphite/water and graphite/ice interfacial systems were first equilibrated on the canonical (NVT) ensemble for 20 ns at 300 and 200 K respectively with an integration timestep of 1 fs. Equilibration was followed by a 100 ns-long production run during which configurations were saved for analysis with an intermittance of 5 ps.

Water molecules were described using the TIP5P water model (Mahoney and Jorgensen, 2000), which reproduces well most thermodynamic properties of both solid and liquid water. For carbon atoms we used a simple generic carbon model from the OPLS potential family (Jorgensen and Tirado-Rives, 1988; Jorgensen et al., 1996) with modified angles and torsions to describe the graphene structure. The carbon-water interactions were modeled by a Lennard-Jones potential between the carbon and the oxygen atom of the water molecule whose parameters ($\epsilon_{OC}^{LJ}$=0.3920 kJmol$^{-1}$ , $\sigma_{OC}^{LJ}$=0.315 ) reproduce the single water molecule-graphite binding energies needed to produce realistic contact angles (Werder et al., 2003) . C-H interactions were ignored. Long range Lennard-Jones interactions were truncated to zero beyond a 1 nm cutoff, while long-range Coulomb interactions were taken into account using the particle mesh Ewald (PME) method (Darden et al., 1993).

## 2.3 Estimation of the interaction energies corresponding to $A_{FHH}$ and $B_{FHH}$

$A_{FHH}$ and $B_{FHH}$ refer to interaction energies between the adsorbent and the first and consecutive adsorbed layers of water as illustrated in Figure 2 (a). For our analysis we defined layer boundaries using fixed cutoff distances determined based on the mass density profile of the ice slab (Figure 2 (b)), which was calculated in rectangular bins parallel to the graphite/ice interface. The layer boundaries were determined at the base of the peaks; the layer thickness is 0.4 nm. This value is consistent with the dimensions of a unit cell of hexagonal ice but does not correspond strictly to a monomolecular layer. The choice is rationalized by the fact that a monomolecular layer is not capable of accommodating an ice unit cell, and the ordering of at least 2 monomolecular layers is needed for ice-like structures to form(Lupi and Molinero, 2014) When calculating the layer-by-layer interaction energies in this work we concentrated on the first four layers of water/ice. Given that the carbon atoms have zero partial charge and the Lennard-Jones parameters of the hydrogen atoms of the TIP5P water model are assumed to be zero, the interaction energy between the graphite slab and the water molecule reduces to the Lennard-Jones interactions between the oxygens of the water molecule (OW) and the carbon atoms (C) of the graphite slab. In this approximation the mean interaction energy per unit area in the $i^{eth}$ adsorbed layer with the slab can be written as:

$$E_{Li} = \frac{4\epsilon_{OC}}{A} \sum_{j=1}^{NW_{Li}} \sum_{k=1}^{NC} \left[ \left( \frac{\sigma_{OC}}{r_{jk}} \right)^{12} - \left( \frac{\sigma_{OC}}{r_{jk}} \right)^{6} \right], \tag{2}$$



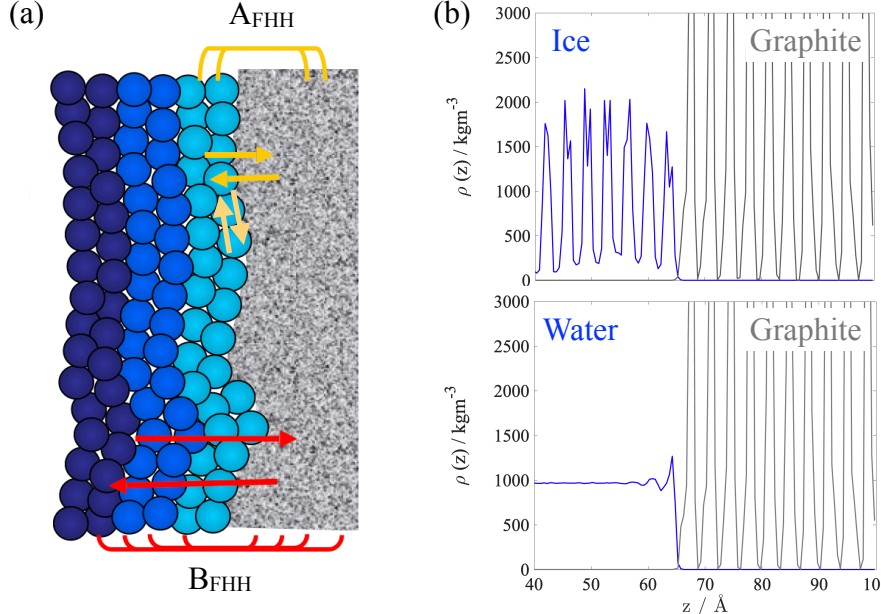

**Figure 2.** a) Schematic representation of the FHH parameters. b) Mass density profiles of the atomistic ice/graphite (top) and liquid water/-graphite (bottom) interfacial systems, shaded areas represent the first four consecutive layers that were considered in the energetics analysis.

where $A$ is the surface area, $NC$ stand for the number of carbon atoms, $NW_{Li}$ is the number of water molecules in the $i^{eth}$ adsorbed layer and $r_{jk}$ are the pairwise distances between the $j^{th}$ oxygen atom in the layer and the $k^{th}$ carbon. The in-layer lateral interaction energies that are included in $A_{FHH}$ correspond to the energy of lateral hydrogen bonds between the water molecules within the layer.

## 3 Results and discussion

### 3.1 Adsorption, deposition freezing and PCF

GCMC/MD trajectories were used to provide a qualitative description of the initial steps of deposition freezing on the surface of our model graphite slab, as well as in a cylindrical and a hemispherical graphitic pore having depths of 2.7 nm and a radii of 2.6 nm. In the following section the freezing curves and the structure of the adsorption layers are analysed as a function of surface geometry to reveal i) the mechanism of deposition freezing, and, ii) any similarities or differences with liquid water adsorption.





### 3.1.1 Adsorption and freezing on non-porous graphite

Figure 3 a) and b) show the time evolution of the number of adsorbed water molecules (adsorption curve, $N_{wat}$) and the fraction

of frozen water molecules (freezing curve) on the non-porous graphite surface. The frozen fraction is calculated as the sum of hexagonal, cubic and interfacial ice divided by the total adsorbed amount. Since before the saturation of the adsorption layer all water molecules added to the system instantaneously adsorbed on the surface, the adsorbed amount was taken to be equal to the total number of water molecules. The number of adsorbed water molecules in Figure 3 starts to increase significantly at around 0.5 ns, after the initial reversible attachment of a handful of individual water molecules directly to the surface.

The steady increase of water molecules begins when additional water molecules attach to pre-existing surface water, which is coherent with the primary mechanism of adsorption of water on graphitic carbon (Popovicheva et. al. (Popovicheva et al., 2004)). During the supercooled liquid adsorption phase pre-critical ice embryos appear in the droplet in a random manner, both in the bulk of the droplet and at the graphite surface. The onset of freezing, i.e. when the number of ice-like water molecules starts increasing linearly, can be observed at $\sim 1.5$ ns on Figure 3 b) (point I.) when the total number of water molecules

reaches $\sim 1000$. The critical nucleus size is $\sim 80$ molecules, determined by directly counting the number of molecules in the largest ice-like cluster at the onset of freezing. The critical cluster - unlike a large fraction of the transient pre-critical embryos - is attached to the graphite surface (Fig 3 c) point I.) In this particular simulation, initially two droplets of similar size develop simultaneously. In a parallel repetition under the same conditions a single droplet forms. Given that the graphite surface is completely homogeneous both chemically and structurally, with no specific adsorption active sites, the number of droplets

forming is determined by stochastic processes. The average contact angle of the droplets ($\theta \sim 90$ °) was estimated from the height ($H$) and radius ($R$) of the droplet as $\theta = tan[R/(H - R)]$, neglecting line tension corrections. This value is merely $4°$ larger than the experimental value for liquid water, suggesting that the dropletwise phase of deposition freezing can be described assuming the same average adsorption layer geometry (droplet shape) as for liquid water adsorption.

Freezing starts in one of the two droplets, and both continue to grow by adsorption until they merge at $\sim 2.1$ ns (point II. in

Figure 3). The merging of the initial droplets does not have an impact on the contact angle. Freezing in the other droplet starts only after merging ($\sim 2.5$ns), however it is initiated from an ice embryo that is distinct from the originally frozen droplet, as seen in the top right panel of Figure 3 c) (snapshot II.) and in the region highlighted by a red circle in Figure 5 b). The fact that this second ice embryo grows starting from the surface suggests that our qualitative conclusions are not influenced by the presence of the ice seed. We also note here that supplementary simulation performed with a MC/MD ratio of 20 (Appendix

A) resulted in the melting of the original ice seed during the first 100 ps of the simulation, followed by the formation of a critical nucleus at the graphite surface at a distinct location. The snapshots in Figure 3 c) suggest that freezing is initiated at the graphene surface and the liquid/vapor interface remains largely liquid-like throughout the dropletwise phase, that is, until complete multilayer surface coverage is reached and the number of water molecules is stabilised in the system. Even in the multilayer phase a non negligible fraction of the liquid/vapor interface exhibits liquid-like structure at full coverage, the frozen

fraction never exceeds 0.7 and non-ordered molecules are concentrated at the droplet surface, as shown in the bottom panels of Figure 3 c). This observation is consistent with a quasi-liquid layer on the free surface of ice and proves that during deposition



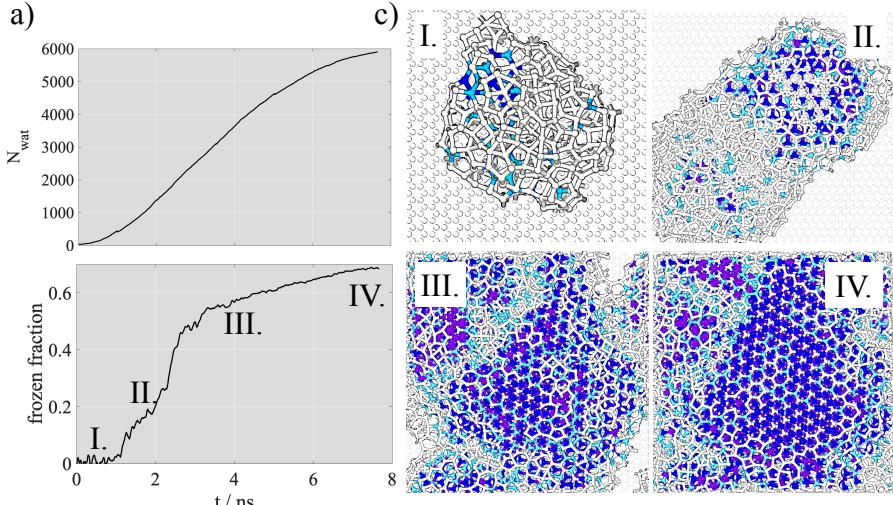

**Figure 3.** (a) Time evolution of the number of water molecules at the flat graphite surface (adsorption curve) (b) The fraction of frozen water molecules as a function of time (freezing curves). (c) Equilibrium snapshots showing the growth and the freezing of the adsorption layer. Liquid-like water molecules are shown in white, hexagonal ice in dark blue, cubic ice in purple and interfacial ice in light blue.

freezing a water molecule attaching to the pre-existing adsorbed droplet or layer has a larger probability to form initial hydrogen bonds with a disordered phase (supercooled liquid) than with ice crystals. Therefore the mechanism (kinetics, energetics) of deposition freezing and liquid water adsorption can be assumed qualitatively very similar - or to first approximation identical.

The ice formed on the surface is stacking disordered in the initial stages, characterized by distinct cubic and hexagonal patches. After droplet merging, cubic and hexagonal patches rearrange into horizontal layers as expected based on previous simulations (Qiu et al., 2018). The cubic to hexagonal ratio in the converged multilayer phase is approximately 0.7.

### 3.1.2 Adsorption and freezing in graphitic nanopores

Freezing curves observed in the two porous systems are compared to the flat surface results in Figure 4 a). Freezing in the

cylindrical pore occurs at 0.8 ns, which is about half the time needed for freezing on a flat surface (∼1.5 ns). Because of this acceleration, freezing is slightly enhanced in the cylindrical pore, but with a freezing curve qualitatively similar to that on the flat surface (i.e. after a linear increase of the frozen fraction between 0.8 and 2 ns), the curve plateaus below 0.7 and ice growth slows down. In the hemispherical pore ice formation is completely suppressed, despite that pore filling follows approximately the same linear pattern (Figure 4 b) as in the case of the cylindrical pore. The frozen fraction remains below

0.1 after the melting of the small ice seed initially present in the pore. This latter finding is in line with the results of Bi et. al. (Bi et al., 2017) who observed a non-monotonous suppression of immersion freezing in graphitic wedges as a function of wedge angle. Looking closely at the structure of the bottom of the hemispherical pore we observe that the radius of the





lowermost part is 0.5 nm, presumably too small to accommodate a critical ice embryo. Additionally, the angle that describes

the aperture of the hemispherical pore (Figure 4 b) is 125 °, which is an angle found to suppress immersion freezing by Bi et.

al. (2017). The interplay of these geometric hindering factors is potentially responsible for the suppression of freezing in the

hemispherical pore. We note that the suppression of freezing despite of the presence of the ice seed evidences that the seed

does not have a significant impact on the qualitative mechanism of ice formation in our simulations. This result is important

because it indicates that ice nucleation depends on the surface geometry, which contradicts the hypothesis that in-pore freezing

occurs homogeneously.

Freezing onset in the cylindrical pore coincides with a filled pore volume fraction of 0.2 and is preceded by the formation

of supercooled liquid nanodroplet until point I., the snapshot of the nanodroplet having the majority of the water molecules

in the disordered phase is shown in point I. of Figure 4 c. The critical nucleus size is ∼80 molecules in this simulation,

which is slightly smaller than the critical nucleus size of the model in homogeneous freezing mode at the homogeneous

freezing temperature (100-200 molecules) (Lupi et al., 2014a), however due to the stochastic nature of ice nucleation this small

difference in the critical nucleus size does not directly suggest a heterogeneous freezing mechanism. The value corresponds to

the critical size observed on the flat surface. The pore filling fraction at freezing onset is shifted to 0.6 for the simulation where

the MC/MD ratio is 20 (Appendix A). In neither case is complete pore filling required for freezing. The plateau in the frozen

fraction (at ∼2 ns) is reached when the filled volume fraction is approximately 0.6, i.e. before complete pore filling. The plateau

is maintained until $V_f/V_{pore}$=1.1, which corresponds to an out of pore nucleation of approximately two molecular layers of

ice. In this initial stage of out-of-pore ice growth the nucleating phase protrudes from the pore assuming flat circular disk shape

up to a height of 2 molecular layers. The ice slab has a strong stacking disorder, with alternating cubic and hexagonal layers

and a cubic to hexagonal ratio of ∼ 1.2.

The surface of this disk is mainly disordered (snapshot II. in Figure 4 c), liquid-like water is concentrated around the three

phase contact line. Adsorption then proceeds by the lateral spread of the disk. In this phase, the attachment of water molecules

occurs mostly around the three phase contact line and results in liquid-like clusters of irregular size and shape around the

cylindrical ice slab protruding from the pore (snapshot III. in Figure 4 c). The accumulation of liquid-like water molecules

is reflected in the freezing curve by a small decrease of the frozen fraction (III. in Figure 4), because the rate of adsorption

around the ice slab responsible for the lateral spread of the droplet exceeds the rate of freezing. In the final phase of freezing,

the spherical cap geometry is recovered, the freezing curve assumes a second linearly increasing segment, with a steepness

however much smaller than that of the first linear segment. This behavior disappears at higher MC/MD ratio, and the shape of

the water protruding from the pore turns out to depend on the extent of freezing when out-of-pore nucleation occurs. The free

surface of the adsorbate remains largely disordered even in this final stage, which is evidenced by the frozen fraction remaining

smaller than 0.7 even at full multilayer coverage.

### 3.1.3   The detailed mechanism of adsorption, pore condensation and freezing

To elucidate the mechanism of deposition freezing in relation to pore condensation and adsorption as underlying basic physical

processes, we need to understand how the adsorbed water/ice builds up on the surface. It is possible to gain the necessary





**Figure 4.** (a) Time evolution of the frozen fraction on the flat surface (gray) and in the cylindrical (teal) and hemispherical (blue) pores (b) Top and side views of the adsobing water at the freezing onset (I.), at initial stage (II.) more advanced stage of out-of-pore nucleation (II.). Water is indicated with ball and stick notation, sticks standing for intermolecular bonds. Carbon atoms in graphitic pore are shown as white balls. Liquid water molecules are shown in white, hexagonal ice in dark blue, cubic ice in purple and interfacial ice in light blue.

information by dividing the adsorbate into layers and observing the growth of the consecutive layers as a function of time. Figure 5 shows the time evolution of the total number of adsorbed (top panel, $N_{tot}$) and frozen molecules (bottom panel, $N_{ice}$) in the first three consecutive adsorbed layers on the flat surface and in the cylindrical pore. The layer boundaries were deter-

mined assuming a layer width of 0.4 nm for all three analysed layers. On the flat surface, layer 2 starts forming when layer 1 contains around 300 water molecules, (corresponding to approximately 17% of saturation in layer 1) as the saturated layer (plateau) contains ∼ 1800 water molecules. Layer 3 starts building up when layer 1 and layer 2 contain ∼ 1000 and ∼ 800 water molecules, corresponding to 56% and 41% saturation respectively. In the cylindrical pore the onset of formation for all





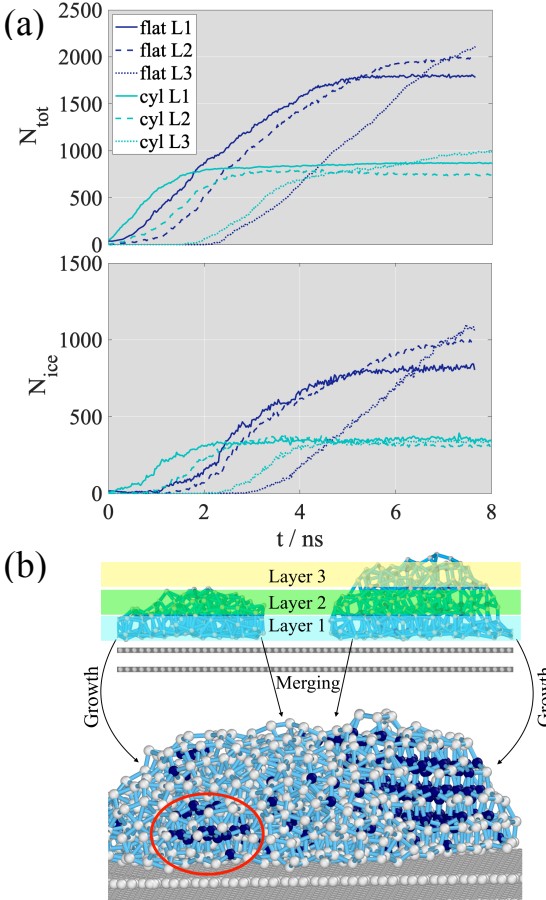

**Figure 5.** The number of water molecules (top panel) and the number of frozen water molecules (bottom panel) in the first three consecutive layers of water (L1: solid line, L2: dashed line, L3: dotted line) on the flat surface (dark blue) and in the cylindrical pore (cyan).

three layers is shifted to the left compared to the flat surface, indicating enhanced adsorption in the pore, but the mechanism of

adsorption is similar to that observed on the non-porous surface. Layer 2 starts forming when layer 1 is 47% saturated, the formation of layer 3 sets on layer 1 is filled and layer 2 is at a 58% saturation level compared to the plateau value that corresponds to the number of water molecules in the completely filled layer. The fact that new layers start forming on unsaturated lower layers proves that adsorption is dropletwise and the continuous multilayer forms as a result of merging of smaller droplets.

Freezing curves in the first three consecutive layers shown the bottom panel of Figure 5 reveal that freezing starts in the

first layer near the surface for both the flat surface and cylindrical pore, the critical nucleus is attached to the surface. The freezing of the consecutive layers is delayed in time, and follows curves that are parallel to the freezing curve in layer 1. On the flat surface, freezing onset in layer 1 and layer 2 coincides approximately with the beginning of the formation of layer 2. The freezing of layer 3 is delayed, onset of freezing is seen when layer 1 and layer 2 are about 40 and 35% frozen relative to



the plateau value of the corresponding layer-by-layer freezing curves. The critical nucleus spreads across the first two layers and grows horizontally before extending to the third layer, but freezing in layer 1 and layer 2 is not complete (does not reached a plateau) before the appearance of ice in the third layer, meaning that what we observe is not a layer-by-layer ice growth, representative of barrierless freezing. In the cylindrical pore, freezing onset is shifted towards the left. Ice appears earlier for each layer than on the flat surface, however the overall freezing mechanism is the same in the presence and the absence of the pore. Alike on the flat surface, layer 1 starts to freeze first, followed by layer 2 and layer 3. The presence of water molecules in the second layer is necessary for freezing but complete surface coverage is not required.

The fact that the presence of a second adsorbed layer is always required for ice nucleation is in agreement with the structure of the critical nucleus found in immersion and contact freezing from molecular dynamics simulations using the same model as in this work (Lupi et al., 2014a). This behaviour shows that freezing is likely heterogeneous despite the low temperature, and requires neither complete pore filling nor a bulk aqueous phase. Note that more than 50% of the molecules in the nanodroplets in which freezing is observed are interfacial and the bulk water density is not expected to be reached in these droplets because of the layering in the water structure invoked by the presence of the solid interface spanning approximately 1.5 nm. A schematic representation of the freezing mechanism on the flat surface consisting of the partial filling of the surface by nanodroplets having 2-4 layers of water and subsequent freezing and merging of the adsorbed droplets is shown in Figure 5 b). We note that the order of freezing and merging cannot be unambiguously determined from these simulations, because of the presence of the ice seed and the inherent stochasticity of ice nucleation. The fact that liquid adsorption always preceeds freezing challenges the classical view on deposition freezing, which states that this mechanism does not involve the liquid phase at all. On the other hand, the similarity between the flat surface and the cylindrical pore together with the faster freezing within the pore suggests that pore condensation enhances freezing but the presence of pores does not seem to be necessary for deposition freezing to occur. Instead adsorption is a necessary prerequisite. The enhancement of freezing in the porous system is due to the faster condensation of water nanodroplets as a result of the inverse Kelvin effect, which makes reduces the time needed to reach the droplet size suitable to accomodate a critical embryo. We hypothetise that this effect of the pores can be large enough to completely suppress freezing on flat surfaces in mixed porous/flat soot samples by competing for the available vapor phase molecules. To consolidate this finding a larger number of parallel observations on more complex surface structures would be needed.

## 3.2 Energetics

For a simple system of two types Lennard-Jones particles $(i,j)$, $i$ representing the adsorbent and $j$ the adsorbate the FHH isotherm can be formulated using the Lennard-Jones parameters (Frenkel, 1947; Halsey, 1948; Hill, 1949):

$$lnS = \frac{-A_{FHH}}{N_d^{B_{FHH}}} = \frac{\pi}{k_B T} \frac{\epsilon_i \sigma_i^6 \rho_i - \epsilon_j \sigma_j^6 \rho_j}{3h^3 N^3}, \tag{3}$$



$\epsilon$, $\sigma$ and $\rho$ are the Lennard-Jones parameters and density of the adsorbent ($i$) and the adsorbate ($j$). $h$ is the adsorbed layer
thickness. From equation 3 we can derive that in the ideal LJ particle case $B_{FHH} = 3$ and a $A_{FHH}$ can be written as:

$$A_{FHH} = -\frac{\pi}{k_B T}\frac{\epsilon_i \sigma_i^6 \rho_i - \epsilon_j \sigma_j^6 \rho_j}{3h^3}. \tag{4}$$

It is in principle possible to estimate $A_{FHH}$ from equation 4 if $h$ is known. For ice we can unambiguously determine $h_{ice}$ from
the base of the first peak of the density profile presented in Figure 2, its average value is 0.4 nm. For water we determined
the layer width as described in Appendix B as 0.38 nm, this agrees well with previous studies that found the interfacial layer
thickness to be nearly identical to that for ice at the free water surface (0.37-0.4 nm) (kal). According to equation 4 for a model
LJ system having two phases whose density and adsorption layer thickness corresponds to that of water and ice, $A_{FHH}^{wat}$=8.4
and $A_{FHH}^{ice}$=7.3. These values are however not representative for realistic water or ice adsorption on graphene for the following
reasons: First, equations 3 and 4 are only valid for layer-by-layer adsorption, whereas water adsorption on graphitic surfaces
is dropletwise, which means that no continuous monolayer is expected before condensation. Second, $A_{FHH}$ accounts also
for the in-layer interactions between the adsorbates, which in a hydrogen bonding liquid as water are orders of magnitude
stronger than simple Lennard-Jones forces. In addition to this, H-bonds also act between the consecutive adsorbed layers,
which causes an apparent strengthening of their interactions with the adsorbent, thus $B_{FHH}$ is also expected to be smaller than
the value predicted by the theory based on pure Lennard-Jones interactions. Therefore instead of comparing theoretical values
of isotherm parameters, we analyse the underlying interaction energies in the following sections to quantitatively assess whether
deposition freezing and droplet nucleation can be described using the same set of adsorption parameters in the framework of
ANT.

### 3.2.1 Comparison of interaction energies

$A_{FHH}$ measures the lateral spread of the adsorption layer and it is made up of the combination of the interaction energy
between the surface and the first adsorbed layer and the energy of the lateral interaction within the first adsorption layer.
Macroscopically, $B_{FHH}$ is a measure of the width of the adsorption layer, on a nanoscale it comprises the interaction energy
between the surface and the second and consecutive adsorption layers. The smaller the value of $B_{FHH}$ the larger the extent of
the interactions.

Figure 6 shows histograms of the layer-by-layer interaction energy per unit area between the solid surface and the first four
molecular layers of water/ice. The interaction energy of both water and ice decays rapidly with the distance from the solid
surface, only the first two molecular layers give a non-negligible contribution to the overall adsorption energy. The interaction
energies between the surface and the further layers are smaller than -1 kJmol$^{-1}$, and inferior to 3k$_B$T/2, that is the kinetic
energy of a molecule at the corresponding temperature. This means that thermal motion alone is enough to overcome those
values of binding energy. The comparison of the layer-by-layer energy distributions obtained for ice and water reveals that
interactions are similar in magnitude, the mean values 37.78 kJmol$^{-1}$ and 39.86 kJmol$^{-1}$ in layer 1 differ within the statistical
error estimated by the standard deviation of the corresponding distributions. In layer 2 the difference between the mean values
(0.8 kJmol$^{-1}$) exceeds the standard deviations, however it is significantly smaller than k$_B$T, the energy of thermal motion in





two dimensions - that is within the layer - at the lower temperature, again rendering thermal motion sufficient to overcome the energy difference. The qualitative similarity of the decay of the interaction energies together with the quantitative similarity between the energy distributions suggest that water and ice adsorption can be described with the same $B_{FHH}$ for graphitic surfaces if only Lennard-Jones type of interactions are present.

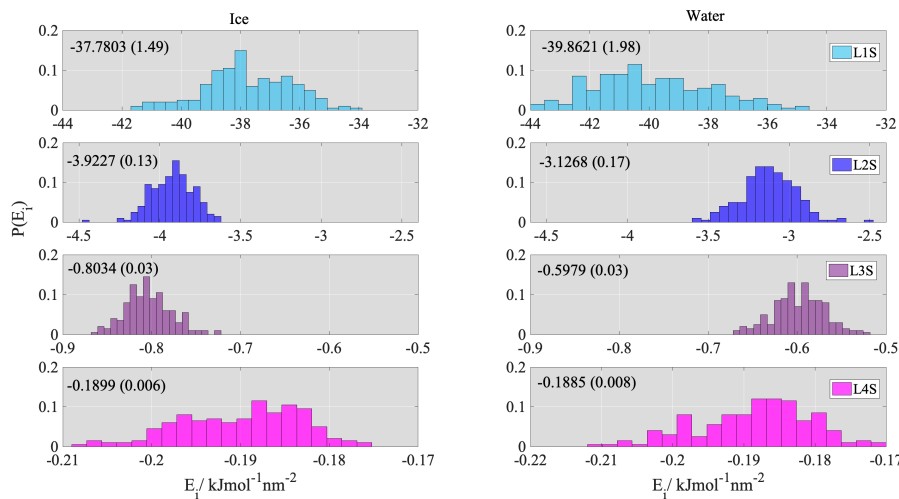

**Figure 6.** Distribution of the interaction energies between the first four adsorbed ice/water layers and the graphite surface.


$A_{FHH}$ also contains lateral interactions between the water molecules, which are governed by hydrogen bonding within the layer. We calculated the average number of hydrogen bonds formed within layer 1 from the simulation using a cutoff of 0.35 nm for donor-acceptor (O...H) distance and 30°for the hydrogen-donor-acceptor (O-H...O) angle. The number of hydrogen bonds per water molecule at the surface is around 2.1 in water and $\sim$3 in ice, the hydrogen bonded network in water appears to be slightly less extended than in ice (Fig 7 (a)). These data are used to estimate the total energy contributions coming from lateral hydrogen bonds in the first layer.

For the TIP5P water model the energy of a hydrogen bond in the bulk liquid phase is -17.7 kJmol$^{-1}$ (Zielkiewicz, 2005). The experimental values for ice range between -18.8 and -13.9 kJmol$^{-1}$ (Nissan, 1956) with an average of 15.5 kJmol$^{-1}$. Given that to the best of our knowledge that hydrogen bond energy data is not available in the literature for ice made up TIP5P water molecules and that the value for liquid water falls within the range of experimental values observed for ice we assumed the per molecule hydrogen bond energy to be the same in both phases. Box plots obtained from the time series of hydrogen bond energies per unit area in layer 1 of water and ice are shown in Figure 7 (b). The average lateral interaction energy in layer 1 of liquid water is -174.7 $\pm$ 12.1 kJmol$^{-1}$nm$^{-2}$ and -181.9 $\pm$ 6.3 kJmol$^{-1}$nm$^{-2}$ in layer 1 of ice. The difference (7.2 kJmol$^{-1}$nm$^{-2}$) is significant compared to the energy of thermal motion, however the boxplots created from the time series of the lateral hydrogen bond energy values clearly indicate that the range of energies excluding outliers (whiskers) for ice



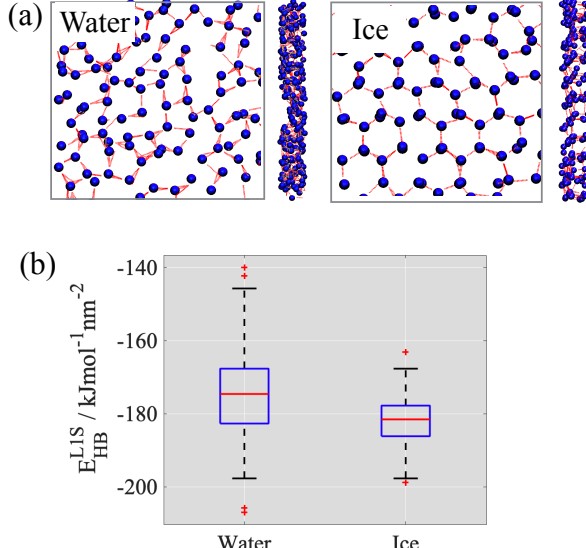

**Figure 7.** (a) Snapshots showing the hydrogen-bonded network in layer 1 of water and ice. Boxplots of the time series of hydrogen bonding energies in the first layer of water and ice.

corresponds to a subset of the range found in water and reveals a significant overlap between the interquantile ranges delimited by the 25 and 75 percentiles (boxes).

## 4 Conclusions

We presented a comprehensive study of deposition freezing on graphitic surfaces in the absence and presence of nanopores with
different geometries with the objective of clarifying the relationship between adsorption, pore condensation and deposition freezing. Our results show that deposition freezing is preceded by the dropletwise adsorption of supercooled liquid water and the free surface of the adsorbed layer remains disordered (i.e., liquid-like) throughout the simulation. This indicates that deposition freezing and water adsorption share similar mechanisms, as water molecules coming from the vapor phase attach to a liquid-like surface. The presence of supercooled liquid nanodroplets that adsorb on the surface before freezing challenges
the traditional view on deposition freezing that it does not involve liquid water: at least a "nanophase" of a few thousand water molecules has to be present before ice nucleation occurs. The freezing observed in the simulation is heterogeneous, as it starts from the first adsorbed layer and does not require the presence of many adsorbed layers nor bulk water. We also showed that pore condensation may accelerate but not cause deposition freezing, as freezing curves on the flat surface and in the cylindrical pore are similar but occur earlier in the latter (likely because of concentrating the adsorbed water due to the inverse Kelvin
effect). The freezing onset preceded the completion of pore filling under both of the two different simulation conditions, one



that enhances freezing and one that results in realistic equilibrium vapor pressure, which is not in line with the hypothesis of PCF freezing stating that pore filling is a prerequisite for ice nucleation. Finally, freezing curves in pores depended on pore geometry: the hemispherical geometry with the small radius enhances adsorption but suppresses freezing, indicating again that pore condensation is not a sufficient nor a necessary condition for deposition freezing to occur.

We also demonstrated the potential that deposition freezing can be described by ANT developed for droplet nucleation, because i) the surface of the adsorbed layer remained liquid-like during the whole simulation, and, ii) we compared the energetics that define the FHH isotherm parameters for water and ice, and found that they are very similar. Future work, which will consist of the direct calculation of the FHH isotherms of water and ice on graphitic surfaces, will focus on further developing ANT to provide a unified parameterization for droplet nucleation and deposition freezing.



## Appendix A: Simulations with MC/MD ratio of 20

An MC/MD ratio of 20 produces realistic equilibrium vapor pressure and this value was also used in the single previous molecular simulation study of deposition freezing (David et al., 2019). The simulations were repeated using this ratio to examine the impact of the choice of this free parameter on the mechanism of deposition freezing on the flat graphitic surface and in the cylindrical pore. The simulations were approximately 2.5 ns-long, sufficient to observe the formation of the critical ice cluster and the initial rapid linear growth of the frozen core. Later phases of the ice nucleation are not expected to depend on the choice of this parameter, since the critical embryo forms at the graphite surface, i.e. embedded in the droplet, thus its growth within the droplet should depend neither on the adsorption of further water molecules on the droplet surface nor on the external vapor pressure.

The left panel of Figure A1 shows freezing curves in the supporting simulations. On the flat surface freezing is preceded by the adsorption of a supercooled liquid droplet which contains approximately 1000 water molecules when the critical ice embryo is observed at approximately 1.5 ns (point If). This is followed by a linear growth of the ice fraction indicating successful nucleation. (point IIf). In the cylindrical pore we also observe the adsoprtion of a supercooled liquid droplet before the formation of the critical cluster and nucleation (Ic.), which occurs slightly earlier (1.2 ns) than on the flat surface. The number of ice-like water molecules remains superior that on the flat surface. While the difference between the two freezing curves is less pronounced at this MC/MD ratio, it is still clear that both the flat and the cylindrical surface nucleate ice and the cylindrical pore has enhanced ice nucleation efficiency compared to the flat surface albeit to a very small extent. Given that freezing was not kinetically favored in this set of simulations, we could observe the melting of the initial ice seed in the early pre-critical stage of the simulation (from 0 to 0.05 ns).

To understand the relationship between pore filling and freezing we show the frozen fraction as function of the pore filling ratio ($V_{filled}$) in the original and the supplementary simulation in the right panel of Figure A1). The freezing onset in the original simulation occurs at $V_{filled} \sim$0.2 and at $V_{filled} \sim$0.6 in the supplementary simulation. Thus the MC/MD ratio strongly impacts the extent to which the pore is filled when nucleation occurs. However, even with the MC/MD ratio that is known to well reproduce realistic pressures the complete filling of the pore is not a necessary condition for freezing.

We assess the impact of the choice of the MC/MD ratio on the qualitative mechanism of adsorption and freezing through snapshots showing the growth and the freezing of the droplet taken from the supplementary simulation. The choice of the MC/MD ratio does not impact most main features of evolution of the droplets and the main conclusion about the freezing processed can be summarized as follows.

1. Freezing is preceded by the adsorption of supercooled droplets. We note that in the supplementary simulation only one such droplet forms, while in the original simulation we observe two smaller droplets. This is however due to the stochasticity of the adsorption process as in a parallel repetition of the simulation with original MC/MD ratio of 5 we also observe a single droplet.





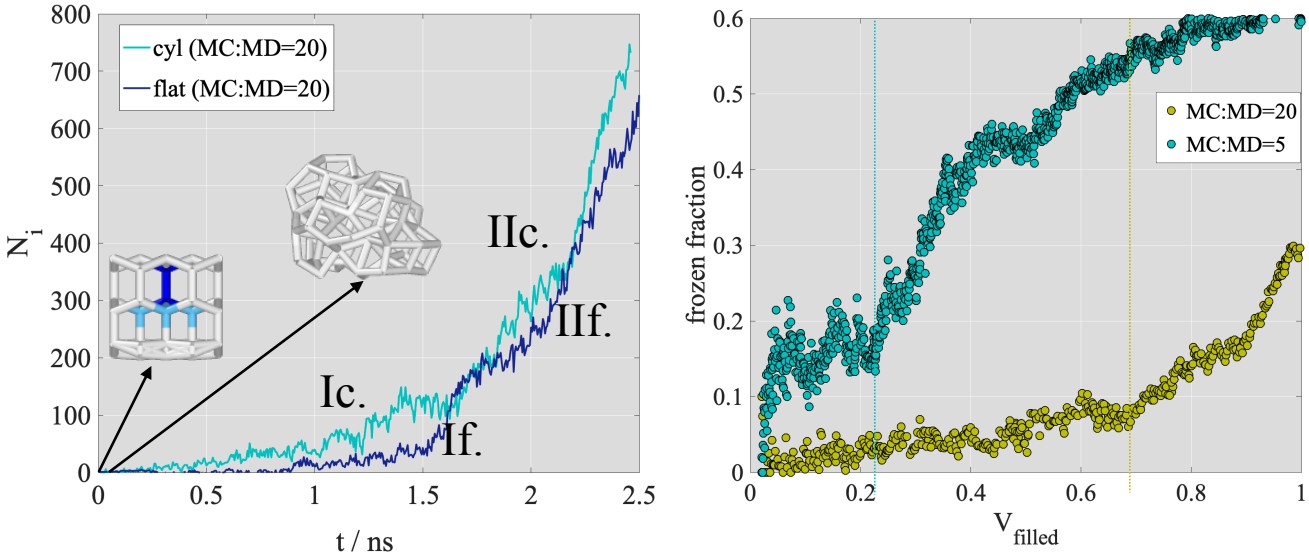

**Figure A1.** (Left panel: freezing curves calculated from the MC/MD=20 simulations. The snapshots in the insets show the initial melting of the ice seed and are taken at t=0 and t=0.05 ns in the cylindrical pore setup. Right panel frozen fraction as a function of the pore filling ratio ($V_{filled}$) for the two types of setups in the cylindrical pore. The dotted vertical lines indicate the freezing onset.

2. Even after the freezing onset have been reached, the droplet surface and in particular the three phase contact line remains liquid like, thus the fact that liquid adsorption and freezing - i.e. the attachment of molecules from the vapor to an existing ice embryo or a liquid droplet - are qualitatively similar.

3. The critical ice embryo forms near the surface, spreads approximately two layers of water and evolves vertically and horizontally, therefore freezing is heterogeneous and the freezing mechanism is in line with the mechanism proposed for immersion and contact freezing involving the same surface and model parameters based on simple MD simulations(Lupi et al., 2014a). Full surface coverage or complete pore filling is not required for the freezing to occur. In case of the cylindrical pore the width of the adsorbed droplet is definitely larger than 2-3 molecular layers on the surface, therefore
it cannot be unambiguously proved that the freezing occurs in the deposition and not immersion/contact mode.

4. The critical ice embryo is distinct from the original ice seed which melts in the beginning of the simulation.

Layer-by-layer analysis of the adsorption and freezing in supplementary simulations is summarized in Figure A3. The top panel shows the number of adsorbed water molecules as a function of time in the first (L1) second (L2) and third (L3) adsorbed layer. Similarly to what has been observed in the original setup the curves suggest that the adsorption follows a dropletwise
mechanism, i.e. a new layer starts building up before the layer below is saturated (the number of adsorbed molecules reaches a plateau). The bottom panel shows the number of frozen molecules in the first three consecutive layers of water as a function of time. Freezing is clearly heterogeneous, i.e. freezing of layer 1, which is in direct contact with the surface starts before that of layer 2 and layer 3. However just like adsorption freezing does not proceed through a layer-by-layer mechanism, which would





**Figure A2.** Top left: side view snapshots showing the evolution of the droplet adsorbed on the flat surface. Botton left: side view snapshots showing of the droplet in the cylindrical pore. The three blue hexagons indicate the position of initial ice seed. Right panel: top view of the critical ice cluster on the flat surface (top) and in the cylindrical pore (bottom). The blue hexagons indicate the place of the original ice seed which melts in the initial few picoseconds of the simulation. White bonds stand for liquid-like water, light blue for interfacial ice, dark blue for hexagonal and purple for cubic ice, carbon atoms are not shown for clarity.

be indicative of a barrierless freezing transition(). On the flat surface the freezing of layer 1 starts when layer 1 , layer 2 and
layer 3 ∼340, ∼250 and ∼160 contain molecules respectively, which is well below the saturation level, in line with the fact
that freezing onset is observed from the simulation in the dropletwise phase. In the cylindrical pore freezing starts when layer
1, layer 2 and layer 3 contain ∼330, ∼270 and ∼230 water molecules, all of them below saturation level. The freezing of layer
2 and layer 3 sets on when layer 1 is a saturated and layer 2 and layer 3 are at an approximately 90% saturation level.



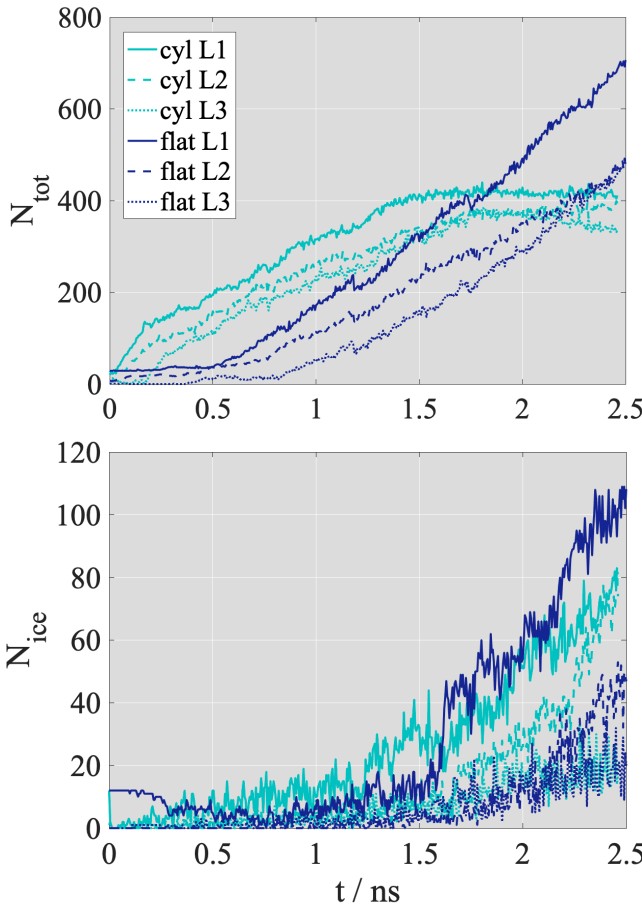

**Figure A3.** The number of water molecules (top panel) and the number of frozen water molecules (bottom panel) in the first three consecutive layers of water (L1: solid line, L2: dashed line, L3: dotted line) on the flat surface (dark blue) and in the cylindrical pore (cyan).

All the above findings are in qualitative agreement with the results of the original simulations, therefore the choice of this
free parameter does not impact main qualitative conclusions. We acknowledge that a more thorough study involving a series of MC/MD ratio values would be needed to assess which is the optimal choice of this parameter that can be used for quantitative simulations including the calculation of nucleation rates.



## Appendix B: Water density profile and interface layer thickness

The number density profile of the first molecular layer of water is shown in Appendix Figure B1 together with the intrinsic number density profile of the bulk aqueous phase. The profiles were calculated using the ITIM (Sega et al., 2018) algorithm.

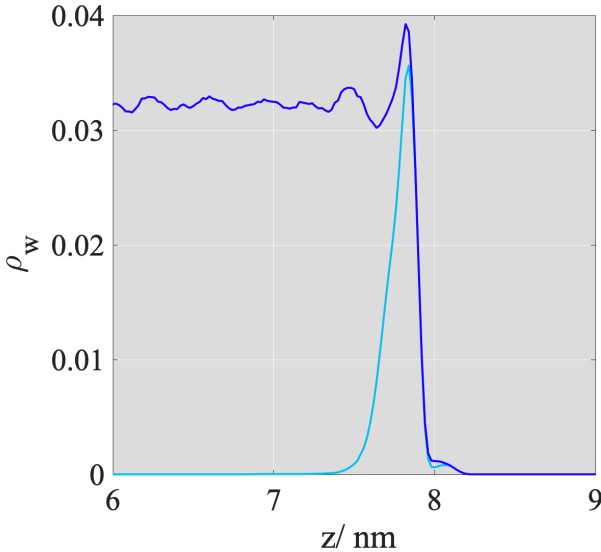

**Figure B1.** (Intrinsic number density profile of the aqueous phase (dark blue) and the first molecular layer of water (light blue).


The width of the first water layer used for the theoretical calculation of the FHH parameters is taken as the width of the first layer profile, which is 0.38 nm at 10% height. (The choice is such to match the calculation of the width of the ice layers and also matches experimental data of 0.37 nm(kal).



*Author contributions.* M.D designed and performed simulations and analysis and wrote the manuscript. A.L. and A.N. contributed to analy-
sis, interpreted results and wrote the manuscript.

*Competing interests.* The authors declare no competing interests.

*Acknowledgements.* A.L. acknowledges support by the Academy of Finland (grant no. 345125 and ACCC Flagship funding, grant no.
337552). A.N. and M.D. acknowledge funding from the Swiss National Science Foundation (Project numbers: CRSK-2_195329 and SNF
200021_169506), project PyroTRACH (ERC-2016-CoG) funded from H2020-EU.1.1. - Excellent Science - European Research Council
(ERC), project ID 726165 and the European Union Horizon 2020 project FORCeS under grant agreement No 821205.



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
