# Peer review of "Deposition freezing, pore condensation freezing and adsorption: three processes one description?"

_EGUsphere, 2023_

## Author Comment (AC1)

**We appreciate the authors' efforts putting this manuscript together, which proposes Adsorption Nucleation Theory (ANT) as a unifying theory of pore condensation and freezing (PCF), adsorption, and deposition nucleation. However, we want to point out that in certain portions of the manuscript, PCF is misrepresented.**

**On lines 76–86, PCF is correctly summarized as involving pore filling followed by immersion or homogeneous freezing of the pore water and ice growth out of the pores. But on the next lines (86–91), it insinuates that the foundation of this framework is based on molecular simulations in David et al. (2019), which is not the case. Instead, PCF describes pore filling with the Kelvin equation, ice nucleation with classical nucleation theory (CNT) parameterizations based on experimentally determined homogeneous ice nucleation rates, and ice growth again based on the Kelvin equation (Marcolli, 2014; 2020; David et al., 2019; 2020; Marcolli et al., 2021). In David et al. (2019), molecular simulations are used to show that (i) ice does not nucleate on the substrate, which was tailored to mimic the silica surface, (ii) ice also does not grow out of a single ice-filled pore with a diameter of 3 nm, but (iii) that ice is growing out of closely-spaced 3 nm pores.**

Thank you for pointing out the inaccuracy in our description of PCF, we will correct this in the revised manuscript.

**As in PCF ice nucleation occurs in pores, it is very different from deposition nucleation and ANT, which both rely on an ice-nucleating surface as the location of ice nucleation. Therefore, the claim to unify PCF and deposition nucleation under a theory based on ANT cannot be kept unless PCF is strongly distorted. For PCF in its right description, the question posed in the title has therefore to be answered with "no". We think a title change is appropriate to reflect this.**

Pores are physical entities, which means that the only difference between the interior pore surface and the exterior surface of an INP is geometry. The basis of ANT is a combination of the FHH adsorption theory and the Kelvin effect. Pore geometry induces an inverse Kelvin effect, which can be accounted for in the ANT framework, as we have shown in Laaksonen and Malila (2021) (subsection 10.7.1). In both cases, our simulations show that ice nucleation proceeds by adsorption of liquid like water onto the surface that then freezes to ice, starting from the surface. We therefore believe that the title of our paper is quite appropriate.

**Moreover, it is incorrect that in David et al. (2019) droplet emulsion experiments were used to show that ice nucleation rates are high enough to occur in the small volumes of pore water (as stated on line 85). Instead, we use slurry experiments (and not emulsion freezing experiments) in David et al. (2020) to show that the pores in the mesoporous silica particles are wide enough to hold ice. Marcolli (2020) discusses in more detail the role of homogeneous nucleation rates. There, it is also shown that the water volume just needs to be large enough to hold the critical ice embryo for water to freeze when temperatures fall below about 230 K.**

We will correct the description on line 85. The pore volume vs. critical cluster size calculated using CNT is at best circumstantial evidence for homogeneous nucleation and does not prove that the freezing inside pores always happens in the "bulk" of the water instead of starting from the pore surface.

**On lines 420–422, the authors write that PCF involves pore filling as a prerequisite for ice nucleation. This statement might be a misinterpretation of David et al. (2019), where ice nucleation in the cylindrical pores of mesoporous silica particles is investigated. The filling of cylin-**

**drical pores is occurring almost instantaneously when RH is above a threshold value. Therefore, these pores are either completely filled or empty. Yet, pores with other shapes like conical pores and wedges fill gradually as discussed in Marcolli (2020). For such pores, ice nucleation occurs when the water volume is large enough to hold a critical ice embryo. Moreover, it occurs irrespective of whether the surface is ice nucleating when temperatures are below about 230 K.**

We will correct the statement on lines 420-422. We agree that at T < 230 K, if the freezing does not start from the interior pore surface, then it will occur homogeneously. However, our simulations indicate that at least in the graphene-water system, freezing starts from surface. Both the original and the control simulations (performed with different MD/MC ratios) clearly show that freezing begins in the layer adjacent to the surface and not in the bulk. Figure 5 and Appendix Figure A3 follow the time evolution of the frozen fraction within the first three layers adjacent to the graphene surface. Before freezing onset transitional ice clusters appear and melt in the bulk of the adsorbed water, while it is only the ice embryo anchored to the surface that can grow beyond the critical size and result in the freezing of the whole adsorbed layer within the simulation time. This is a qualitative indicator that heterogeneous freezing at the surface is energetically favorable compared to the homogeneous process. Additionally, the fact that the hemispherical pore does not nucleate ice despite of the condensation rate being similar as observed for the cylindrical pore indicates the importance of the surface. The volume of the two pores is comparable and both can accommodate critical embryos of homogeneous freezing in the bulk however, it is only the cylindrical pore, with the bottom of the pore being smooth and having an area that is sufficient for the formation of a critical embryo that promotes freezing, indicating that heterogeneous freezing is favured over homogeneous freezing in the studied systems.

Moreover we point out that the statement "the pore is either filled or empty" only makes sense if the temporal resolution of the observation is known. The temporal resolution of the simulations is on the order of picoseconds, therefore they are able to resolve process that appears to be instantaneous in experiments.

**PCF was introduced to explain measured ice nucleation data as e.g. in Marcolli (2014) and in Marcolli et al. (2021). In the latter, the requirements for ice nucleation on soot particles were established taking the primary particle size, overlap, soot contact angle, and soot aggregate size into account. This soot PCF framework can explain why some types of soot nucleate ice while others do not. Specifically, it predicts that soot particles with a contact angle of 90° do not nucleate ice below water saturation because, according to the Kelvin equation, there is no capillary condensation in pores for contact angles of 90° or higher. Conversely, the graphitic surface in the present study shows an unrealistically high water adsorption of several monolayers for $RH_w$ < 100 % despite its contact angle of 90°. A reason for this strong water adsorption might be that the simulation was carried out at a supersaturation of 300 % $RH_i$ or $S_i$ = 300 % (lines 145–146). A high saturation ratio of 250 % $RH_i$ was also used for the simulation with mW water shown in Fig. 3 of David et al. (2019) to speed up the simulation. Nevertheless, the simulation in David et al. does not indicate significant water adsorption on the flat silica surface although its contact angle of 64° is clearly below the one of the graphitic surface and the simulation time was much longer (300 ns compared to 10 ns in the present study).**

We would like to point out that the capillary condensation equations used for calculating pore filling may be inaccurate at small pore sizes (Kruc and Jaroniec, 1997). As noted above, we have shown

that the ANT equations can be used to predict pore filling RH (Laaksonen and Malila, 2021), accounting explicitly for the molecular interactions (described using the FHH parameters) between the water meniscus and the pore walls. With large pores, our approach gives identical results to the capillary condensation calculations, but not when the pore radii are only a few nm. For example, assuming the FHH parameters for carbon of A = 12. B = 1.93 (Laaksonen et al., 2020) and contact angle of 90 degrees, we calculate that a hemispherical pore with a radius of 2.6 nm is filled at an RH of 89% at room temperature.

We acknowledge that the supersaturation of 300% used in the simulation is exaggerated and does not model atmospheric conditions. Similarly to the simulations shown in David.et. al. (2019), where two supersaturation values are used (250% and 300%), we adapt this value to increase the speed of the simulation by increasing the driving force. We stress that we cannot convert the instantaneous vapor pressures (numbers of molecule in the vapor phase) to relative humidity in this simulations because, exactly due to the exaggerated driving force, the vapor phase is not expected to be equilibrium with the adsorbed layer at all steps of the simulation. This means we cannot unambiguously determine the structure of the adsorbed layer as a function of equilibrium relative humidity, therefore it is not meaningful to state that we reach several molecular layers thick adsorbed layer at RH<100%.

While acknowledging the above caveat of using exaggerated vapor pressure to speed up the simulations, we do not believe that it leads to unphysical adsorption and that adsorption on a molecularly flat graphite surface is not expected to happen. First of all, experimental evidence from HREEL spectroscopy of water adsorbed on monolayer graphene shows that at room temperature not only water adsorbs on the surface but even chemisorption and C-H bond formation are expected (Politano et. al., 2011). An SFG study that finds pre-ordering of water near graphite and relates it with the potential IN activity of graphene (Singla et. al., 2017). Quantum chemical calculations also yield negative adsorption energies for the water/graphene system (Ma et. al. 2011). Atomistic molecular dynamics simulations of water adsorbed on the surface of graphene(Gordillo&Martí, 2008) and graphite (Gordillo&Martí, 2008) predict adsorption energies on the order of -30 kJ/mol.

In David et. al. (2019) the authors relate the IN activity of the silica surface to the extent of permelting: "*Recent experiments ([59]) quantitatively confirm the simulations prediction ([30]) of the fraction of water that is premelted in silica pores as a function of temperature. We have demonstrated, using thermodynamics and nucleation theory, that surfaces that induce premelting (as is the case of silica) cannot heterogeneously nucleate ice from the liquid phase ([31])*." (David et. al. (2019)). We note that while amorphous silica might induce premelting, graphene has an ordering effect on the surface water molecules which makes permelting unlikely. This has been seen from sum frequency generation spectroscopy (Singla et. al. ,2017) as well as atomistic molecular dynamics simulations (Gordillo&Martí, 2008), and quantum chemistry (Leenaerts et. al., 2009, Sanfelix et. al., 2003 ), the latter supporting that hexagonal ordered monolayers are the most stable configuration at the graphene surface. Therefore the fact the amorphous silica does not nucleate ice due to pre-melting does not rule out the that graphene will adsorb water and promote heterogeneous ice nucleation. And the observation that in our simulations, freezing starts from the surface, already indicates that graphene does not induce premelting.

**Moreover, experiments with the non-porous particles do not show any deposition nucleation (David et al., 2019). The question therefore arises why the water adsorption is so high in the simulations with a graphite slab (non-porous) and monatomic water despite the high contact angle of 90°.**

The experiments of DeMott et al. (1999), Mahrt et al. (2018) and Nichman et al. (2019) show that non-porous soot nucleates ice at rather high ice supersaturation, but still below water saturation at temperatures below 230 K.

**Another point that sheds doubt on the meaning and relevance of the simulation results shown in Lbadaoui-Darvas et al. is that the graphitic surface proved to be an efficient ice-nucleating agent in immersion mode in molecular simulations with mW water (Lupi and Molinero, 2014; Lupi et al., 2014). Yet, experiments have shown that soot is a poor INP in immersion mode or even ice nucleation inactive (Hoose and Möhler, 2012; Kanji et al. 2020). We wonder why the authors chose a graphitic surface for their study, although the mW water model is known to overpredict the ice nucleation activity of the graphitic surface in immersion mode (Qiu et al., 2018). The high water adsorption together with the false prediction of IN activity may explain the ability of the mW water to nucleate ice on a flat graphitic surface. Yet, these simulations do not represent real physical processes occurring in or on soot particles.**

As acknowledged before in the response as well as in the manuscript, we are aware of the caveats of the water model as well as of the lack of testing and validation for adsorption studies. The overprediction of freezing on graphene using the mW potential is due to the model's inability to reproduce the contact angle of water and the free energy of the ice-graphite interface simultaneously. This caveat is intrinsic to the resolution of the model and arises from the lack of rotational degrees of freedom, therefore only atomistic potentials could resolve the issue.

Unfortunately, the only MD program package (LAMMPS) that can be used to perform hybrid GCMC/MD simulations does not allow for parallelisation of this particular simulation type if the adsorbate is a truly polyatomic molecule. Atomistic simulations in this case would come at absurdly high computational cost due to the fact that freezing is a rare event and it can be observed from unbiased simulations only at the best IN surfaces (AgI, kaolinite). The potential parameters used in the simulation were chosen because they represent the best available compromise in terms of computational cost and accuracy.

 In any case, we do not expect to obtain quantitative information relevant to real graphitic surfaces from the coarse grained simulations. Even if the water model overpredicts freezing on graphite, our aim in this manuscript is not to make quantitative predictions of ice nucleation on porous and non-porous graphitic surfaces, but rather to reveal the mechanisms of ice nucleation and their similarities/dissimilarities in the two cases. We believe that for this purpose, the systems we have chosen for our simulations are completely adequate. It is also clarified in the text that the surfaces used in the GCMC/MD simulations do not represent real graphene and we refrain from using these simulations to predict any quantitative data. Quantitative data (interaction energies) are only obtained from atomistic MD simulations.

REFERENCES

P. J. DeMott, Y. Chen, S. M. Kreidenweis, D. C. Rogers, D. E. Sherman. Ice formation by black carbon particles. Geophys. Res. Lett. 26, 2429-2432 (1999).

M. Kruk, M. Jaroniec, Application of large pore MCM-41 molecular sieves to improve pore size analysis using nitrogen adsorption measurements, Langmuir 13 (1997) 6267–6273.

Laaksonen, A., Malila, J., and Nenes, A.: Heterogeneous nucleation of water vapor on different types of black carbon particles, Atmos. Chem. Phys., 20, 13579–13589, https://doi.org/10.5194/acp-20-13579-2020, 2020.

Laaksonen, A. and Malila, J.: Nucleation of water, Elsevier, 2021.

Mahrt, F., Marcolli, C., David, R. O., Grönquist, P., Barthazy Meier, E. J., Lohmann, U., and Kanji, Z. A.: Ice nucleation abilities of soot particles determined with the Horizontal Ice Nucleation Chamber, Atmos. Chem. Phys., 18, 13363–13392, https://doi.org/10.5194/acp-18-13363-2018, 2018.

Nichman, L., Wolf, M., Davidovits, P., Onasch, T. B., Zhang, Y., Worsnop, D. R., Bhandari, J., Mazzoleni, C., and Cziczo, D. J.: Laboratory study of the heterogeneous ice nucleation on black-carbon-containing aerosol, Atmos. Chem. Phys., 19, 12175–12194, https://doi.org/10.5194/acp-19-12175-2019, 2019.

Politano, A. R. Marino, V. Formoso, G. Chiarello; Water adsorption on graphene/Pt(111) at room temperature: A vibrational investigation. *AIP Advances* 1 December 2011; 1 (4): 042130. https://doi.org/10.1063/1.3660325

Saranshu Singla, Emmanuel Anim-Danso, Ahmad E. Islam, Yen Ngo, Steve S. Kim, Rajesh R. Naik, and Ali Dhinojwala ACS Nano **2017** 11 (5), 4899-4906 DOI: 10.1021/acsnano.7b01499

Ma, J., Michaelides, A., Alfe, D., Schimka, L., Kresse, G., & Wang, E. (2011). Adsorption and diffusion of water on graphene from first principles. *Physical Review B*, *84*(3), 033402.

Gordillo, M. C., & Marti, J. (2008). Structure of water adsorbed on a single graphene sheet. *Physical Review B*, *78*(7), 075432.

Gordillo, M. C., & Martí, J. (2010). Effect of surface roughness on the static and dynamic properties of water adsorbed on graphene. *The Journal of Physical Chemistry B*, *114*(13), 4583-4589.

Leenaerts, O., Partoens, B., & Peeters, F. M. (2009). Water on graphene: Hydrophobicity and dipole moment using density functional theory. *Physical Review B*, *79*(23), 235440.

Sanfelix, P. C., Holloway, S., Kolasinski, K. W., & Darling, G. R. (2003). The structure of water on the (0001) surface of graphite. *Surface science*, *532*, 166-172.

---

## Author Comment (AC2)

We thank both reviewers for the thorough revision of the paper and for the useful suggestions. Below we discuss the changes made in the manuscript in response to the reviewer comments.

**Answers to the comments of reviewer 1.**

*"I think it should be stressed that the terminology/definition of deposition ice nucleation is historically macroscopically defined (before the application of in situ microscopy and MD simulations). For many current experimental techniques, this still has validity. Though on a molecular level this may not be true. Following the conventional definition, it is "deposition ice nucleation" and not "deposition freezing". Liquid (macroscopic) water freezes, but deposition ice nucleation does not involve (following convention) bulk liquid water. If the authors by purpose mix these two definitions and generate a novel terminology, since they observe deposition ice nucleation to originate from a liquid-like water cluster, then this has to be discussed. However, this seems not to be the case since this term is readily used. Also, I would not challenge the conventional definition based on one simulation study only. Hence, "deposition freezing" should be exchanged for deposition ice nucleation throughout the manuscript."*

We agree that using a more conservative and coherent terminology is beneficial for the paper and we have changed all instances of *"deposition freezing"* to *"deposition ice nucleation"*

*"The other issue regarding terminology is to call the water clusters "droplets", "dropletwise", etc. I see that the authors struggle with this issue as well, trying also "nanodroplets" or "nanophase". In this community droplets are usually defined to be 10s of micron in size. The "nanodroplets" forming inside the pore are about 2 nm or smaller. Typically, we call those entities clusters. It may not even be clear if this cluster size possesses bulk-liquid water properties (surface tension, etc.)? I am also not entirely sure how to name those condensed nanometer-sized islands of water but naming those "droplets" is unfortunate and ambiguous. Liquid-like or ice-like water clusters may be an idea. Maybe "nanodroplets" works to convey the idea but I feel this is not ideal either."*

[Figure]

Figure 1: The radial mass density distribution of a simulated water droplet (from unpublished work of the authors).

We agree that the water aggregates observed before condensation are presumably too small to be called droplets and in order to clearly differentiate form other uses of the word droplet in aerosol science (e.g. cloud droplet) we adapt the term *"nanodroplet"*, however *"dropletwise adsorption"* is a standard term used for non filmwise adsorption, and has been used to describe ANT based droplet nucleation on non completely wettable surfaces in previous publications ( Laaksonen, Malila and Nenes 2020: https://acp.copernicus.org/preprints/acp-2020-202/acp-2020-202.pdf), therefore we prefer to keep it for the description of the adsorption mechanism and the structure of the adsorbed layer. However, we have added a phrase to define *"dropletwise adsorption"* as *"adsorption that proceeds through the formation of aqueous nanodroplets on the surface that merge into multilayer film at higher supersaturations."* We refrain from using the term *"clusters"* in order to avoid confusion with the terminology of new particle formation that typically refers to non-stabilised molecular aggregates with diameters lower than 1 nm

as clusters. We justify the choice of using the term *"nanodroplets"* by sharing results from atomistic simulations of water nanodroplets on flat graphene having around 1000 molecules, that correspond approximately to the size of the droplets in the dropletwise phase of these simulations. The percentage of surface molecules determined by selective surface analysis (PYTIM) reaches 50% of the total number of molecules but the density within the droplet reaches its bulk value, as seen in the figure above. This analysis suggests that the water droplets are clearly different from macroscopic droplets ased on the surface to bulk ratio, but they cannot be called clusters, which have a well defined bulk that reaches the density of the corresponding macroscopic phase.

*"From the abstract and introduction, one would expect some analysis using ANT, i.e., deriving ice nucleation rates, etc. However, this study makes the case that ANT can be applied to deposition ice nucleation based on the simulation results. As written, this fact may not be so clear, and the overall confidence is only supported by this study looking at one idealized substrate. Maybe in some instances the text could convey a more exploratory study. I do not disagree with the authors; I suggest being a bit more conservative. Especially when reading the model methods. Many caveats are discussed (which I appreciate, and this does not minimize the impact of the study) but it feels a bit counter (i.e., weaker) to the introduction. This is maybe something the authors could consider."*

We have considered the concerns raised by the reviewer and made the following modifications to clarify that the presented results are to be viewed as a case study that evidences the potential to extend ANT for ice and droplet nucleation:

1. Abstract: "We put forward the plausibility of ....." was changed as follows: "Based on the results of our case study, we put forward the plausibility

2. Abstract: "but the input parameters are also potentially transferable across phase states of the nucleating phase." was changed to "but the input parameters are also potentially transferable across phase states of the nucleating phase at least for the case of the graphite/water model system." to clarify that our study does not necessarily imply general extension of ANT to deposition ice nucleation.

3. Introduction: "In this work we address all of the above three crucial points using atomistic and coarse-grained molecular simulations." was changed to "In this work we address all of the above three crucial points using atomistic and coarse-grained molecular simulations, with aims i) of providing novel insights into the so much debated mechanism of deposition freezing; and ii) of demonstrating that adsorption and deposition ice nucleation are sufficiently similar in terms of mechanism and thermodynamics for ANT to be used as a collective framework to describe both of them."

4. Conclusions: "We also demonstrated the potential that deposition freezing can be described by ANT developed for droplet nucleation" was changed to "We also demonstrated the potential that deposition ice nucleation on the model graphite/water system can be described by ANT with the parameters developed for droplet nucleation" to convey the idea that at this stage results are system specific.

**Specific comments:**

*Line 22: The authors could cite here the recent review by (Knopf and Alpert, 2023).*

The requested citation was added

*Line 30: I doubt that (DeMott et al., 2010) discuss in detail nucleation theory and rates relevant for this paper and they do not discuss specifically deposition ice nucleation. Other literature would be needed in this place.*

The reference was corrected, the revised manuscript refers to recent reviews on ice nucleation: Hoose et. al. 2012, Kanji et. al. 2017, Welti et. al. 2014

*Line 37: "...adsorbed water can exist...."*

The sentence was rephrased according to the suggestion

*Line 40: At this point it is not clear what you mean by "whereas other locations that collect pre-critical clusters might have an opposite effect." Why do they have an opposing effect?*

The reason for this is that certain active sites might collect water due to the inverse Kelvin effect but have geometries that do not allow the formation of pre-critical ice clusters, as it was seen for the hemispherical pore geometry in our simulations. To clarify this in the text we have added the following phrase: *"whereas other locations that collect pre-critical clusters might have an opposite effect, for instance because their geometry disfavors the formation of pre-critical ice embryos (Bi et. al. 2017)."*

*Line 47: Here it suddenly switches to immersion freezing. I recommend keeping the focus on deposition ice nucleation throughout the introduction. Also, I am not sure if I agree with this statement. When CNT is expressed in terms of water activity, intrinsic parameters like contact angle, interfaces, etc. are considered. See, e.g., the work by Knopf and Barahona groups. In fact (Knopf and Alpert, 2023) show that deposition ice nucleation may be well described using water activity as for the case of homogeneous ice nucleation and immersion freezing.*

We understand that the molecular simulations cited in this section were conducted in immersion mode, however we cannot refer to work on deposition ice nucleation because ours is the first such study, safe for the simulations presented in David. et. al. 2019. We clarify this in the revised version and we argue that pre-critical enhancement or suppression of ice formation is presumably equally important in any ice nucleation mode. Further we clarify - both here and in the revised manuscript - that while surface specific properties can be treated by different versions of CNT (e.g.: the soccerball modell by Niedermeyer or the activity based formulation), even these cannot take into account pre-critical enhancement or suppression or freezing, because the original theory considers ice nucleation as a single step process with no assumption about surface-water interactions before the critical nucleus formation.

*Line 55-57: Missing words, empty brackets?*

Missing citations were added.

*Line 73-75: As mentioned above, considering water activity in CNT descriptions might account for these issues (Knopf and Alpert, 2023; Koop et al., 2000; Barahona, 2015, 2014; Knopf and Alpert, 2023).*

Citations for models treating surface water interactions were added, but we maintain that CNT is inherently unable to account for pre-critical stages of ice nucleation.

*Line 81: Period missing?*

This is addressed in the next point.

*Line 81-86: A long sentence, maybe too long. Also, this statement is too general. Careful literature review will show that there are several studies (some employ nanoscale resolution) which do not corroborate PCF occurring in observed deposition ice nucleation experiments, e.g. (Wang et al., 2016). I would avoid "in reality" and write "…'freezing' could be pore condensation…".*

The sentence was divided into smaller sentences and rephrased to avoid declaring PCF as the only plausible mechanism of deposition freezing, what is implied by the expression "in reality".

*Line 112: It is crucial to conduct atomistic and coarse-grained molecular simulations as discussed in (Knopf and Alpert, 2023) and shown in (Roudsari et al., 2022). They can yield different results while atomistic simulations are likely the preferred method, if feasible.*

The requested sentence was added to the manuscript.

*Line 127: What do you mean by energetic background? This is not a thermodynamic expression. Maybe just state the parameters you assess?*

The expression has been changed to *"interaction energies that drive adsorption and deposition ice nucleation"*

*Line 146-147: Could you please elaborate here. The target vapor pressure corresponds to the adsorption layer structure"? At such high vapor pressure, one would have multiple layers of water? I assume, this is what you want?"*

The statement has been rephrased as follows: *"The target vapor pressure is not to be confused with the instantaneous vapor pressure in the simulation box, it is strictly the vapor pressure that corresponds to the value expected in the fully converged simulation (i.e. at multilayer coverage)."*

*Line 159: What do you mean by "deterministic dynamics"?*

By deterministic dynamics, we mean standard Newtonian dynamics of classical simulations. We stress the importance of this condition by adding the following note in the manuscript: *"Note that it is important to ensure that the spatial dynamics in the system are not driven by stochastic changes introduced by the Monte Carlo sampling, otherwise it is impossible to estimate time dependent properties from the simulation."*

**Answers to the comments of reviewer 2.**

**Specific comments:**

*The atomistic simulations of the graphite/water interface, used to determine the FHH parameters employ the TIP5P water potential. While TIP5P overall predicts structural and thermodynamic properties of water and ice quite well, it does underestimate the density difference between water and ice Ih [see e.g. Vega et al., J. Phys.: Cond. Matter 7, S3283–S3288 (2005)]. Do you believe that using a water model yielding more realistic differences in density between liquid and ice phases, in conjunction with the fixed definition of the layer thickness, could lead to systematic differences in the average interaction energies obtained, in addition to possible differences in interaction energies resulting from the different water model?*

The difference between the bulk density of water and Ih is 0.015 gcm$^{-3}$ and 0.07 gcm$^{-3}$ for TIP5P and TIP4P/ice. This appears to be large enough to non-negligibly alter interaction energies. However, given that these are differences between the mean bulk densities, they can be used as proxies to predict the dependence of interaction energies on the water model choice only in the bulk, where the impact of the vicinity of the surfaces on the density profiles especially of the liquid phase is negligible. In our systems, seen from the density profile of liquid water (Figure B1.) this is expected to occur below layer 3. As we state in the manuscript, in the particular case of graphene/water systems *"The interaction energy of both water and ice decays rapidly with the distance from the solid surface, only the first two molecular layers give a non-negligible contribution to the overall adsorption energy."* therefore we do not expect that the bulk density difference can be directly related to potential differences in the interaction energy across different water models.

It could on the other hand be useful for future studies to compare the density profiles of different water models near the surface and provide depth-dependent estimate of the sensitivity of interaction energies to the choice of the water model. However, such a study requires a number of simulations and potentially the use of different surfaces, and it goes far beyond the scope of this work.

*I would encourage the authors to review their variable names and formulae. Throughout the manuscript, sub- or superscripts in variables that do not denote indices are set in italics (e.g. $A_{FHH}$, $k_B$, $\sigma_{OC}$, $V_{filled}$, $N_{ice}$, etc.). These sub- and superscripts should be changed to roman font. In addition, several variable names appear unnecessarily long or convoluted to me (e.g., in eq. 2, the number of carbon atoms "NC" could be $_1N_{C]}$ and the number of water molecules in i-th adsorbed layer, "$NW_{Li}$" could simply be "$N_i$", so that the mean interaction energy per unit area in the i-th adsorbed layer could simply be denoted "$E_i$".*

We have reviewed the variable names according to these suggestions, however NC and NW were kept. Instances of denoting the number of water molecules as Ntot or Nwat were also changed to NW.

**Technical corrections:** We appreciate for pointing out technical issues. We addressed each of them as requested in the manuscript. For those where we feel that additional explanation of the changes made in the manuscript is need, we provide a detailed explanation below the comment.

*l.14: not only the extension . . . is possible -> not only is the extension . . . possible*
*l. 39: clusters , -> clusters,*
*l.55: implemented regional -> implemented into regional*
*l.57: two broken references*

*l.69: ...with direct links to the molecular-scale interactions...*
*l.139: ...a graphite slab with a hemispherical or cylindrical pore consisting...*
*l.164: numercially->numerically*
*l.171: remove comma after carbon*
*l.176: remove spurious "because"*
*l.178: any estimating -> estimating any*
*l.194: units missing for $\sigma_{OC}^{LJ}$ (I assume nm)*

The correct unit (nm) was added.

*l.206: space missing before reference.*
*Figure 2 caption: describe arrows and label water and graphite in panel (a). Shaded areas mentioned in the caption not visible in panel (b).*

In panel a) the arrows were labelled $E_1$, $E_{HB}$ and $E_{2/3}$ to indicate the interaction energy component that they represent, graphite and water were labelled. In panel b) the shaded areas are now shown with the same color coding as used in Figure 6.

*l.230: increase of adsorbed water molecules*
*l.231: re-format reference*
*Figure 3: "b)" is missing in the figure. A period is missing at the end of the first sentence in the caption. Carbon atoms are not mentioned in the caption.*

The figure was corrected, panel b) is now correctly labelled and we indicate that carbon atoms (alike liquid-like water molecules) are shown in white.

*l.299: steepness -> slope?*
*Figure 4: "teal" and "blue" curves are very hard to distinguish; please change one of the two colors! "gray" curve appears black to me, especially as the graph has a gray background. Period at the end of the first sentence of the caption is missing. "adsobing" -> adsorbing. "out-of-pore nucleation" -> ice growing out of the pore?*

The colors were corrected, now the results for the hemispherical pore are shown in yellow colour. $V_{filled}$ in the axis label of panel b) was changed to $V_f$ consistently with the overall simplification of the variable names. Out-of-pore nucleation was not changed, as out-of-pore nucleation is an established term (Page and Sear PRL, 2006).

*Figure 5: panels (a) and (b) in the actual figure are not references in the caption. Instead, the caption references a "top" and "bottom" panel, which are both part of panel (a) in the actual figure. The content of panel (b) is not explained at all in the figure caption, but it is referenced in the main text...*

The caption was corrected and $N_{tot}$ in the axis label of panel a) was changed to NW for consistency

*l.327: shifted towards the left -> shifted to smaller time values?*
*l.345 remove "makes"*
*l.355 "and a $A_{FHH}$" -> and $A_{FHH}$*
*l.360 broken reference*
*l.381: missing math environment $3k_BT/2$*
*l.385/Fig 6: in the text, $E_i$ has the unit of energy (kJ/mol), in the figure the unit of energy/area (kJ/mol/nm$^2$). Please fix this.*

Correctly, the interaction energies calculated are normalised by the surface area, the mistake was corrected in the text.

*l.386: missing math environment $k_BT$*
*Figure 6: caption: add explanation that "LnS" denotes the interation between the surface and the n-th adsorbed*

*layer in the graphs. add information that the means and standard deviations(?) (in parantheses) of the distributions are also shown as insets on the graphs of the distributions. Check whether the unit of energy/surface area is the correct one for $E_i$ (see l.385 in the main text)*

L1S, L2S... were changed to L1, L2... for consistency and the revised caption explains that they refer to layer 1, layer 2 etc. We also describe the meaning of the insets: mean interaction energy and standard deviation (in brackets). Energy units were corrected in the manuscript text.

*l.393: missing space after 30º*
*l.399: ice made up TIP5P water molecules -> the ice phases of the TIP5P water model?*
*l.401: box plots obtained from the time series. . . -> box plots of the distribution. . .*
*Figure 7: caption: box plots of the time series. . . -> box plots of the distribution. . .*
*l.425: "the potential that deposition freezing can be described" -> "that deposition freezing can/could be described"*